

# GBP2 as a potential prognostic biomarker in pancreatic adenocarcinoma

Bo Liu[1,2,3,*], Rongfei Huang[4,*], Tingting Fu[5], Ping He[1,2], Chengyou Du[3], Wei Zhou[6], Ke Xu[7] and Tao Ren[7]

[1] Department of Hepatobiliary Surgery, Pidu District People's Hospital of Chengdu, Chengdu, China
[2] Department of Hepatobiliary Surgery, The Third Affiliated Hospital of Chengdu Medical College, Chengdu, China
[3] Department of Hepatobiliary Surgery, The First Affiliated Hospital of Chongqing Medical University, Chongqing, China
[4] Department of Pathology, Clinical Medical College and The First Affiliated Hospital of Chengdu Medical College, Chengdu, China
[5] Department of Nosocomial Infection Control, The Third Affiliated Hospital of Chengdu Medical College, Chengdu, China
[6] Department of Radiology, Clinical Medical College and The First Affiliated Hospital of Chengdu Medical College, Chengdu, China
[7] Department of Oncology, Clinical Medical College and The First Affiliated Hospital of Chengdu Medical College, Chengdu, China
[*] These authors contributed equally to this work.

## ABSTRACT

**Background**. Pancreatic adenocarcinoma (PAAD) is a disease with atypical symptoms, an unfavorable response to therapy, and a poor outcome. Abnormal guanylate-binding proteins (GBPs) play an important role in the host's defense against viral infection and may be related to carcinogenesis. In this study, we sought to determine the relationship between GBP2 expression and phenotype in patients with PAAD and explored the possible underlying biological mechanism.

**Method**. We analyzed the expression of GBP2 in PAAD tissues using a multiple gene expression database and a cohort of 42 PAAD patients. We evaluated GBP2's prognostic value using Kaplan–Meier analysis and the Cox regression model. GO and KEGG enrichment analysis, co-expression analysis, and GSEA were performed to illustrate the possible underlying biological mechanism. CIBERSORT and the relative expression of immune checkpoints were used to estimate the relationship between GBP2 expression and tumor immunology.

**Result**. GBP2 was remarkably overexpressed in PAAD tissue. The overexpression of GBP2 was correlated with an advanced T stage and poor overall survival (OS) and GBP2 expression was an independent risk factor for OS in PAAD patients. Functional analysis demonstrated that positively co-expressed genes of GBP2 were closely associated with pathways in cancer and the NOD-like receptor signaling pathway. Most of the characteristic immune checkpoints, including PDCD1, PDCDL1, CTLA4, CD80, TIGIT, LAG3, IDO2, and VISTA, were significantly expressed in the high-GBP2 expression group compared with the low-GBP2 expression group.

**Conclusion**. GBP2 acted as a potential prognostic biomarker and was associated with immune infiltration and the expression of immune checkpoints in PAAD.

**Subjects** Bioinformatics, Diabetes and Endocrinology, Gastroenterology and Hepatology, Oncology

Corresponding authors
Ke Xu, xuke@cmc.edu.cn
Tao Ren, rentao509@outlook.com

**Keywords** GBP2, Pancreatic adenocarcinoma, Prognostic biomarker, Immune infiltrating

## INTRODUCTION

Pancreatic adenocarcinoma (PAAD) is a highly malignant cancer with a poor prognosis. It is the fourteenth most common cancer but is the seventh leading cause of death in cancer patients (*Bray et al., 2018*). PAAD does not produce consistent clinical symptoms in its early stage and by the time a diagnosis is obtained, approximately 80% of patients have lost the opportunity for surgical resection. The 5-year overall survival (OS) rate is less than 6%. With surgical resection, the 5-year survival rate of patients is less than 25% (*Kamisawa et al., 2016*; *Ying et al., 2016*). The etiology and pathogenesis of PAAD have not been well-studied. A long-term history of smoking, advanced age, a high-fat diet, being overweight, and chronic pancreatitis are possible non-hereditary risk factors for PAAD. Heredity is also a high-risk factor for PAAD. *Rahib et al. (2014)* predicted that PAAD would become the second leading cause of death from malignant tumors by 2030. Recent studied have determined the molecular characteristics of PAAD and further study of its pathogenesis and etiology will assist in the discovery of advanced treatment strategies and more effective biomarkers.

Guanylate-binding proteins (GBPs) are members of the superfamily of interferon-inducible GTPases (molecular weight: 65–67 kDa). Seven human-derived GBPs have been identified to date (*Kresse et al., 2008*; *Man et al., 2017*). GBPs play an important role in a host's defense against viral infections and have a wide range of antiviral properties including the human immunodeficiency virus, hepatitis C, classical swine fever, Zika, measles, and influenza A (*Braun et al., 2019*; *Itsui et al., 2006*; *Krapp et al., 2016*; *Li et al., 2016*; *Yu et al., 2020a*). Moreover, dysregulation of GBPs plays key roles in carcinogenesis. GBP3 promotes the proliferation of glioma cells by regulating the SQSTM1-Erk1/2 signal cascade (*Xu et al., 2018*). *Yamakita et al. (2019)* used migration and wound healing tests to confirm that GBP1 may enhance the aggressiveness of lung adenocarcinoma by promoting cell movement. However, GBP1 inhibits the proliferation, migration, and invasion of CRC cells by mediating the anti-tumor effect of IFN-$\gamma$ (*Britzen-Laurent et al., 2013*). Low GBP6 expression has been confirmed in tongue squamous cell carcinoma and is related to the poor prognosis of these patients (*Liu et al., 2020*). There are no reports on the role of GBP-2 in PAAD to date.

We explored the expression of GBP2 in several cancers, including PAAD, using The Cancer Genome Atlas (TCGA) and GEO databases. We further investigated the prognosis value of GBP2 and the potential biological function of GBP2 in PAAD using multi-dimensional analysis. The relationship between GBP2 and tumor immunity was evaluated to elucidate the role of GBP2 in the PAAD tumor environment. Our results may indicate a novel prognostic biomarker and potential therapeutic target for PAAD.

## MATERIALS & METHODS

### Investigation of GBP2 expression

Five microarray datasets including 281 tumor samples and 122 nontumor tissues were obtained from GEO (Table S1). Transcript data and clinical information for PAAD were downloaded from the TCGA-PAAD dataset. The different expression levels of GBP2 in each GEO dataset were analyzed in the R platform. The expression of GBP2 in various type of cancers was investigated using gene expression profiling interactive analysis (GEPIA) (*Tang et al., 2017*) (http://gepia.cancer-pku.cn/).

### Tissue sample and patient information acquisition

A cohort of 42 patients with PAAD who underwent surgical treatment in The First Affiliated Hospital of Chengdu Medical College was included in our study, which was conducted from January 2014 to December 2020. Formalin-fixed paraffin embedded (FFPE) tissue samples were selected. Each case consisted of tumor samples and matched normal tissue samples that were adjacent to the cancerous cells. Forty-two patients' clinical data was collected. This included: gender, age, differentiation, TNM stage, overall survival and survival status (Table 1). The study was approved by the ethics committee of The First Affiliated Hospital of Chengdu Medical College (2020CYFYIRB-BA-1200) and all subjects or an immediate family member signed an informed consent to participate in this study.

### Immunohistochemistry

The tissue samples were obtained and treated by conventional slicing, baking, dewaxing and hydration. We then repaired antigens, used serum to block non-specific proteins, and added rabbit anti-human GBP2 antibody (11854-1-AP, Proteintech) overnight at 4 °C. Goat anti-rabbit IgG (kit-5030, MXB Biotechnologles) was added and incubated at 37 °C. The sections were counterstained with hematoxylin after the treatment with diaminobenzidine (DAB). After dehydration, transparence and sealing treatment, they were observed under microscope. Ten visual fields were randomly selected from each section, the H score was calculated according to the staining intensity and the number of cells, and the mean value was taken. The H score = (percentage of cells of weak intensity × 1) + (percentage of cells of moderate intensity × 2) + (percentage of cells of strong intensity × 3).

### Correlation of GBP2 expression and clinical characteristics and prognosis

The relationship between the mRNA level of GBP2 and the clinical parameters in PAAD patients (including age, gender, tumor stage, node stage, grade and disease stage) were analyzed using the clinical data of TCGA PAAD patients. The *SuvivalROC* (*Heagerty, Lumley & Pepe, 2000*) package was used to generate the area under the curve (AUC) of 3-year survival prediction for GBP2 and the Youden index was calculated as the cut-off point for dividing the patients of TCGA-PAAD into high-GBP2 and low-GBP2 expression groups. Subsequently, we explored the prognostic value of GBP2 by conducting Kaplan–Meier (K–M) plots, and a log-rank test in the TCGA dataset and our cohort.

**Table 1  Clinical information of the study population.**

|  | TCGA | Our cohort |
|---|---|---|
| **Sample size** | 170 | 42 |
| **Age (mean ± SD)** | 64.45 (10.83) | 55.17 (10.59) |
| **Gender (%)** | | |
| Female | 78 (45.9) | 17 (40.5) |
| Male | 92 (54.1) | 25 (59.5) |
| **Grade (%)** | | |
| G1+G2 | 119 (70.0) | 36 (85.7) |
| G3+G4 | 49 (28.8) | 6 (14.3) |
| NA | 2 (1.2) | – |
| **T Stage (%)** | | |
| T1+T2 | 27 (15.9) | 17 (40.5) |
| T3+T4 | 141 (82.9) | 25 (59.5) |
| NA | 2 (1.2) | – |
| **N Stage (%)** | | |
| N0 | 47 (27.6) | 20 (47.6) |
| N+ | 118 (69.4) | 22 (52.4) |
| NA | 5 (2.9) | – |
| **TNM Stage_AJCC(%)** | | |
| I+II | 160 (94.1) | 29 (69.0) |
| III+IV | 8 (4.7) | 13 (31.0) |
| NA | 2 (1.2) | – |
| **Event (%)** | | |
| Alive | 80 (47.1) | 14 (33.3) |
| Dead | 90 (52.9) | 28 (66.7) |
| **Median OS (month)** | 15.35 | 12.50 |

## Independence of GBP2 in predicting survival

Univariate and multivariate Cox regression analyses were applied to assess the independence of GBP2 for predicting OS in PAAD patients. We analyzed the gene expression of GBP2 and the clinical pathological parameters, including age, gender, tumor stage, node stage, grade, and the disease stage.

## Function and KEGG enrichment analysis

We screened out the co-expressed genes of GBP2 using Linkedomics (*Vasaikar et al., 2018*). GO and KEGG analysis were applied for the positively co-expressed genes of GBP2. The Pearson correlation coefficient >0.5 and $P < 0.05$ were used to define the co-expressed genes of GBP2. GSEA was performed to elucidate the different biological processes and signaling pathways between the high- and low-expression groups of GBP2 using the *Clusterprofiler* (*Yu et al., 2012*) package in R platform.

## Association of GBP2 expression with immune cell infiltration

CIBERSORT (*Newman et al., 2019*) is a deconvolution algorithm based on gene expression to describe the composition of complex tissues. LM22 is a gene signature consisting of

547 genes and is used to distinguish between 22 human immune cell subtypes. We used CIBERSORT to estimate the portions of 22 human immune cell subtypes and compared the composition of the high-GBP2 and low-GBP2 expression groups.

### Differential expression of immune checkpoints between high-GBP2 and low-GBP2 expression group

Blocking immune checkpoints (ICs) (*Salmaninejad et al., 2019*) has shown promising results in treating several cancers. There is a difference in expression between most characteristic ICs in high-GBP2 and low-GBP2 expression groups, including PDCD1, PDCDL1, CTLA4, CD80, TIGIT, LAG3, IDO2 and VISTA.

### Statistical analysis

We performed a Kaplan–Meier curves analysis and log-rank test using the *survival* package. We conducted time-dependent ROC curve analysis using the *survivalROC* package. Univariate and multivariate Cox regression were utilized to determine independent factors for OS. A Pearson correlation coefficient of >0.5 and $P < 0.05$ were used as the criteria to define the co-expressed genes of GBP2. We compared the different expression of the immune checkpoints between the high-GBP2 and low-GBP2 expression groups using the Wilcox test. Results with $P$ value <0.05 indicated statistical significance.

## RESULTS

### Overexpression of GBP2 in PAAD

GBP2 expression was elevated in various types of cancers including PAAD (Fig. 1A). The data illustrated in Fig. 1B revealed that GBP's mRNA expression was remarkably higher in PAAD tissues compared with adjacent normal tissues. We further verified GBP2 expression in the TCGA dataset and GTEx project using GEPIA, and generated a consistent result with the five microarray datasets (Fig. 1C).

The immunohistochemical detection of tissue samples from our cohort of 42 patients with PAAD showed that GBP2 was highly expressed in the cytoplasm of PAAD cells, and was weakly expressed in some adjacent pancreatic acini and small pancreatic duct cells. However, GBP2 was not expressed in normal pancreatic acini and islet cells, but was weakly expressed in the cytoplasm of normal pancreatic small duct cells (Fig. 1D). Supportively, there were significant differences in the expression of GBP2 between cancerous tissues and their adjacent normal tissues as determined by a quantitative analysis of the H score (Fig. 1E).

### Overexpression of GBP2 correlates with advanced T stage and poor prognosis

We analyzed the relationship between GBP2 expression and clinical characteristics and found that, the expression of GBP2 was significantly associated with an advanced T stage (Fig. 2A, Fig. S1). Subsequently, we explored the association between the expression of GBP2 and prognostic outcome by conducting KM survival curves. The AUC of 3-year survival prediction of GBP2 was 0.7 and the Youden index was 5.7 (Fig. 2B), and this result was used to categorize the patients into a high-GBP2 expression group and a low-GBP2

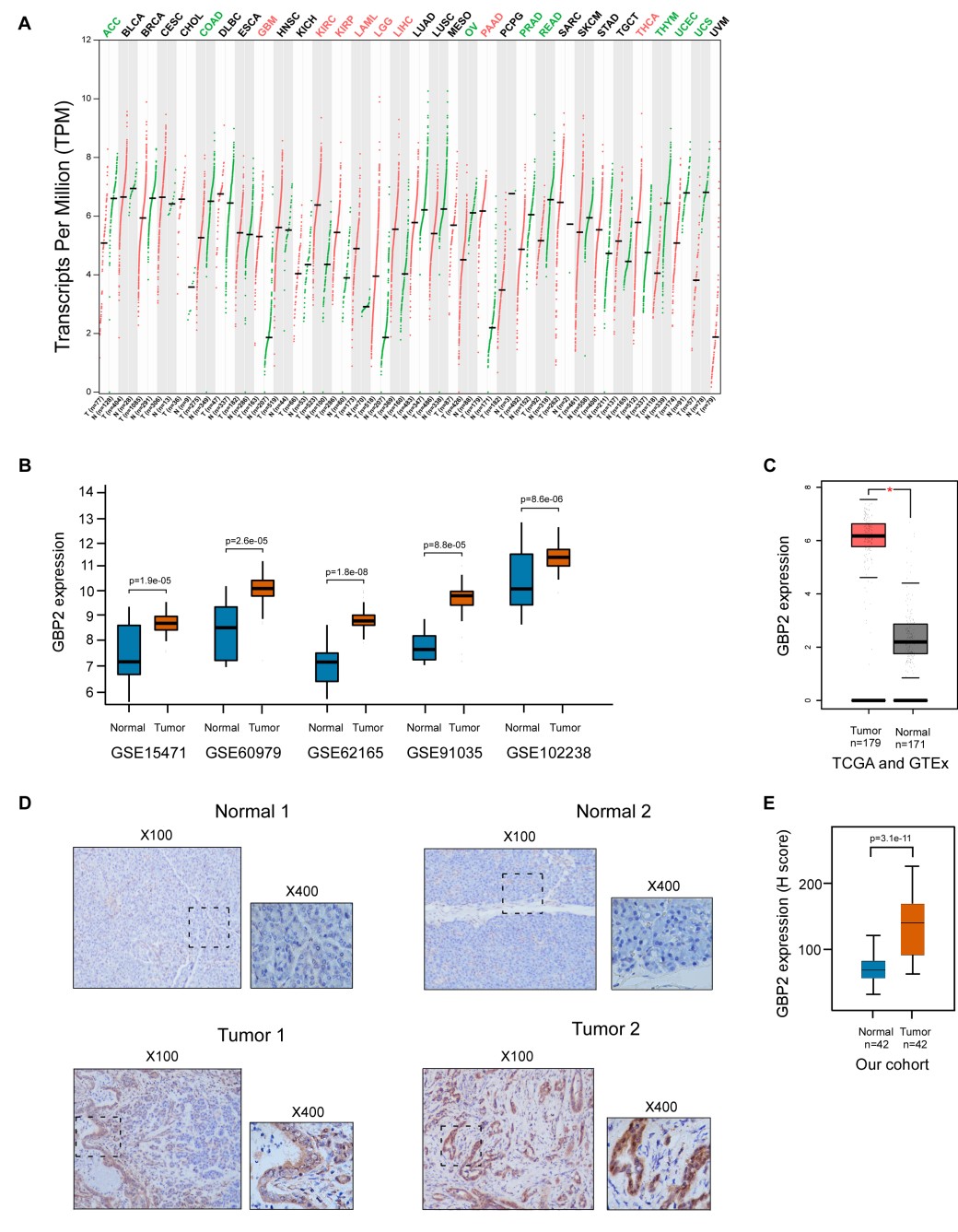

**Figure 1** **Overexpression of GBP2 in PAAD.** (A) The expression of GBP2 in various type of cancers analyzed by using GEPIA online tool. Red represents higher expression in the tumor tissue than in the normal tissue. Green represents higher expression in the normal tissue than in the tumor tissue. Black represents no significant difference between the two types of tissues. (B) The GBP2 expression of PAAD tissues and normal tissues from 5 GEO datasets. (C) The GBP2 expression of PAAD tissues and normal tissues from TCGA database. (D) Immunohistochemical detection of tissue samples from 42 patients. (E) The expression of GBP2 between cancer and adjacent normal tissues from 42 patients by quantitative analysis of H score.

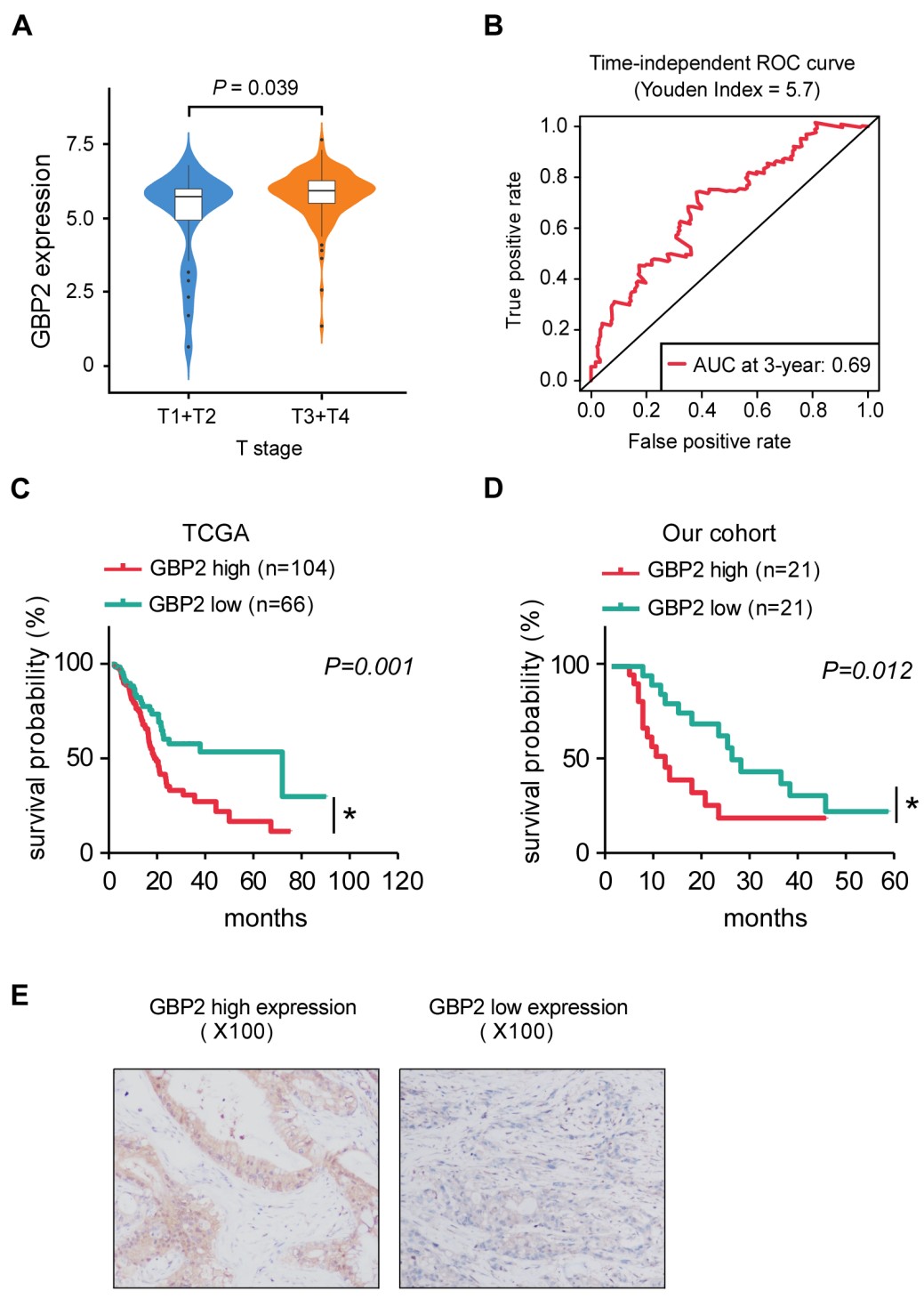

**Figure 2** **The relationship between GBP2 expression and the clinical data.** (A) The relationship between GBP2 expression and T stage. (B) AUC of 3-year survival prediction for GBP2 expression. (C) Kaplan–Meier plots of the high- and low- GBP2 expression PAAD patients from TCGA database. (D) Kaplan–Meier plots of the high- and low- GBP2 expression PAAD patients from our cohort. (E) The representative diagrams of immunohistochemistry for PAAD tissues corresponding to high and low expression of GBP2.

expression group. The overexpression of GBP2 had a significantly shorter OS compared with the low-GBP2 expression group in the TCGA dataset according to the KM curves analysis (Fig. 2C). The 42 PAAD patients included in our cohort were divided into high- and low- GBP2 expression groups by their median of H-score. We also found that the PAAD patients with high GBP2 expression had a poor prognosis (Figs. 2D and 2E). Thus, our results suggested that the expression of GBP2 was markedly associated with a poor prognosis in PAAD patients.

## Independence of GBP2 in predicting survival

Univariate and multivariate Cox regression analysis were conducted to assess the independent prognostic value of the expression of GBP2. Univariate Cox analysis suggested that the expression of GBP2 and several clinical parameters, including age, tumor stage, node stage, and histologic grade, were associated with poor OS in PAAD patients. Multivariate Cox regression analysis indicated that only age and GBP2 expression were independent variables associated with the prognosis of PAAD patients (Fig. 3).

## Function and KEGG enrichment analysis

A total of 222 positively co-expressed genes of GBP2 were analyzed using GO and KEGG analysis (Fig. S2). The significant GO terms for biological processes were associated with the response to interferon-gamma and the cellular response to interferon-gamma (Fig. 4A and Table S2). KEGG pathway analysis indicated that pathways in cancer, including the NOD-like receptor signaling pathway, and salmonella infection, pathogenic *Escherichia coli* infection and apoptosis were the highly enriched pathways (Fig. 4B and Table S3). GSEA revealed the markedly different signaling pathways, including the chemokine signaling pathway, cytokine-cytokine receptor interaction, focal adhesion, pathways in cancer, and regulation of actin cytoskeleton (Figs. 4C–4D and Table S4). These were primarily correlated with carcinogenesis, invasion, and the immune microenvironment of tumor cells.

## GBP2 expression was correlated with immune infiltrating and the expression of ICs

We estimated the differences in immune cell infiltration between the high- and low-GBP2 expression groups by applying the CIBERSORT algorithm. Figure 5A shows the landscape of 22 subpopulations of immune cells in PAAD patients. Patients with high-GBP2 expression had a significantly higher proportion of active CD4 memory T cells, resting Dendritic cells, and M1 macrophages (Fig. 5B). Furthermore, the characteristic ICs, including PDCD1, PDCDL1, CTLA4, CD80, TIGIT, LAG3, IDO2, and VISTA, were more significantly expressed in the high-GBP2 expression group compared with the low-GBP2 expression group (Figs. 6A–6H).

## DISCUSSION

Progress has been made in the molecular research of PAAD, however, the mortality rate has not been significantly reduced due to late diagnoses, early metastasis, and the insensitivity of the disease to chemotherapy or radiation therapy. PAAD has the worst

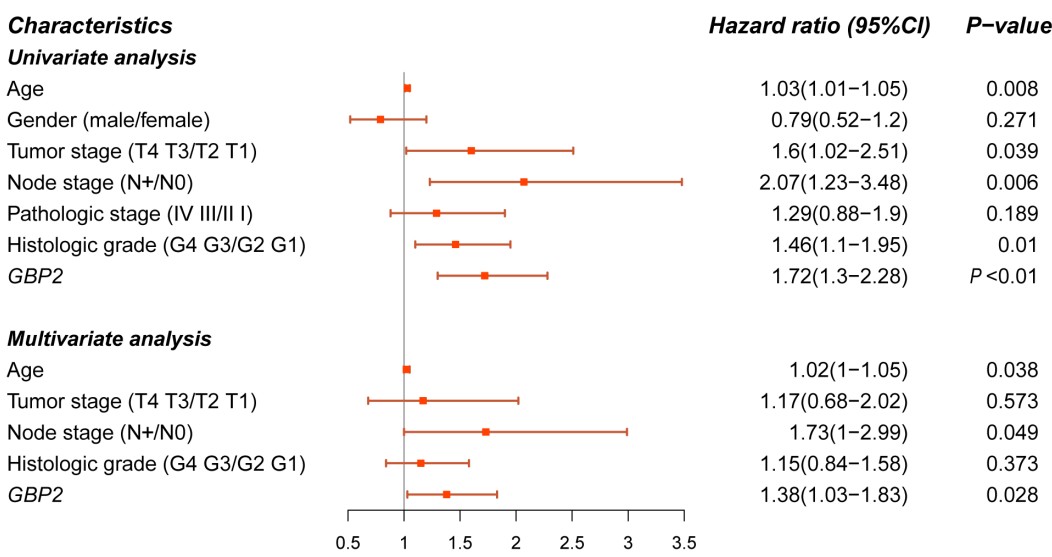

| Characteristics | Hazard ratio (95%CI) | P−value |
|---|---|---|
| **Univariate analysis** | | |
| Age | 1.03(1.01−1.05) | 0.008 |
| Gender (male/female) | 0.79(0.52−1.2) | 0.271 |
| Tumor stage (T4 T3/T2 T1) | 1.6(1.02−2.51) | 0.039 |
| Node stage (N+/N0) | 2.07(1.23−3.48) | 0.006 |
| Pathologic stage (IV III/II I) | 1.29(0.88−1.9) | 0.189 |
| Histologic grade (G4 G3/G2 G1) | 1.46(1.1−1.95) | 0.01 |
| GBP2 | 1.72(1.3−2.28) | P <0.01 |
| **Multivariate analysis** | | |
| Age | 1.02(1−1.05) | 0.038 |
| Tumor stage (T4 T3/T2 T1) | 1.17(0.68−2.02) | 0.573 |
| Node stage (N+/N0) | 1.73(1−2.99) | 0.049 |
| Histologic grade (G4 G3/G2 G1) | 1.15(0.84−1.58) | 0.373 |
| GBP2 | 1.38(1.03−1.83) | 0.028 |

**Figure 3** Univariate and multivariate Cox regression for screening the prognostic factors of PAAD.

prognosis among all common solid malignancies. Therefore, an early diagnosis and reliable prognostic biomarkers are essential to improve the prognosis of PAAD. GBPs are involved in the regulation of intracellular immunity and basic physiological processes, including the proliferation and migration of endothelial cells (*Guenzi et al., 2001*; *Weinlander et al., 2008*). GBP2 is a member of the GTPase superfamily and is important for protective immunity against microorganisms and viral pathogens (*Vestal & Jeyaratnam, 2011*). Recent studies have revealed the role of GBPs in carcinogenesis and have found that GBP2 inhibits the growth of colorectal cancer cells by interfering with WNT signal transduction (*Wang et al., 2020*). The expression of GBP2 in esophageal squamous cell carcinoma (SCC) is significantly higher than that in adjacent normal epithelium, and may be used as an important biomarker for esophageal SCC (*Guimaraes et al., 2009*). Furthermore, GBP2 is highly upregulated in human glioblastoma and enhances the invasive ability of glioblastoma via the GBP-2/Stat3/FN1 signal cascade (*Yu et al., 2020b*). *Godoy et al. (2014)* found that the high expression of GBP2 was associated with a better prognosis in breast cancer, which may be due to the fact that GBP2 represents an effective marker of the T cell response. GBP2 can prevent the translocation of dynamin-related protein 1 (Drp1) from the cytoplasm to the mitochondria, thus weakening the Drp1-dependent mitochondrial fission and the invasion of breast cancer cells (*Zhang et al., 2017*). However, the function and mechanism of GBP2 in the development of PAAD is still unclear. Our findings may help understand the pathological role of GBP2 in the growth, invasion, and metastasis of PAAD cells and assist in the identification of new diagnostic and prognostic marker of PAAD.

In the present study, overexpression of GBP2 was significantly associated with a poor outcome in PAAD patients. The time-dependent ROC curve suggested that the expression of GBP2 could accurately predict the OS. According to the results of univariate and multivariate Cox regression analysis, the expression of GBP2 was an independent

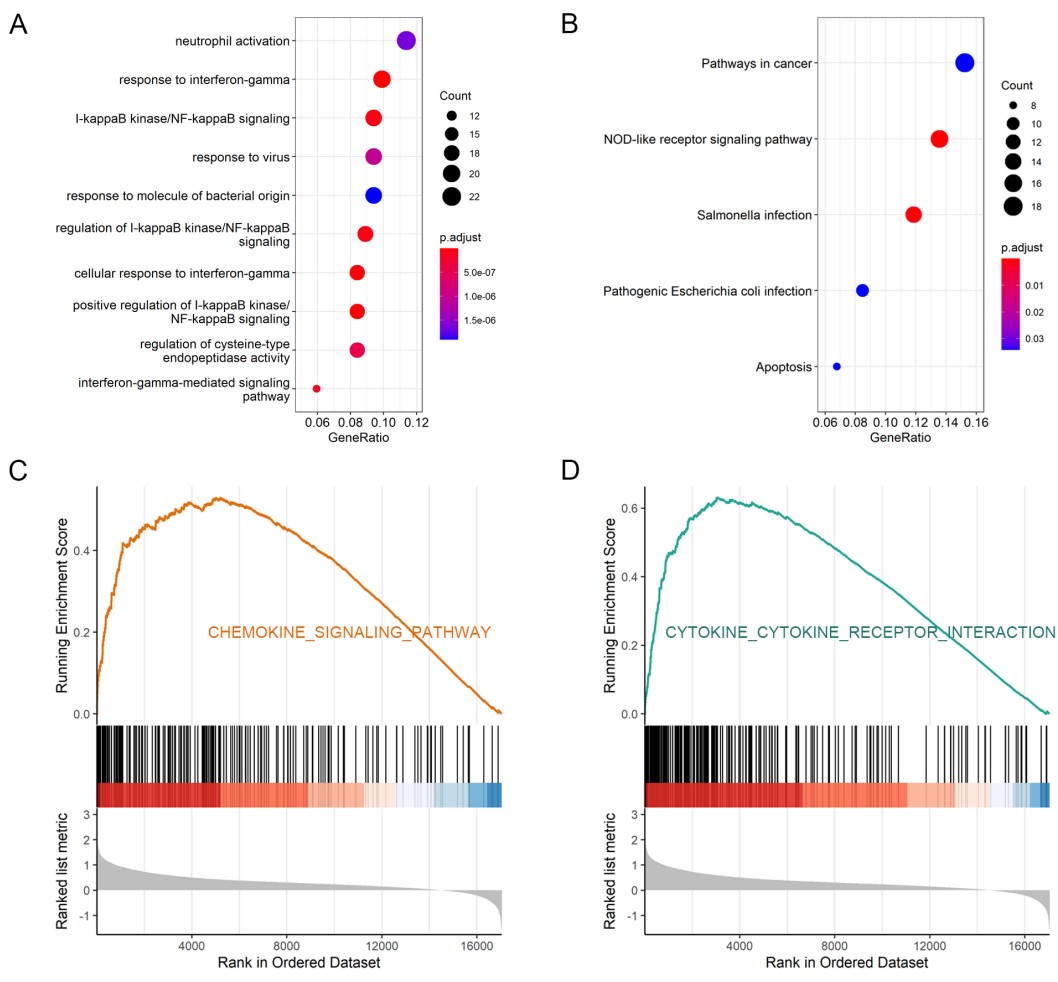

**Figure 4** **Function and KEGG enrichment analysis.** (A) Go enrichment analysis by inputting the 222 positively co-expressed genes of GBP2. (B) KEGG enrichment analysis by analyzing these positively co-expressed genes of GBP2. (C, D) The significantly enriched pathway analyzed by GSEA.

prognostic factor for OS indicating that GBP2 may be used as a prognostic marker for PAAD. The co-expressed genes of GBP2 were mainly associated with the immune response for GO terms related to biological processes. The GBP2-related signaling pathway mainly correlated with tumorigenesis, invasion, and the immune microenvironment of tumor cells. Subsequently, we investigated the tumor immune infiltrating cells in patients with PAAD. The results indicated that overexpressed GBP2 were infiltrated by significantly more activated CD4 memory T cells, resting Dendritic cells, and M1 Macrophages. The infiltration of these immune cells was highly related to the tumor immune response. Activated CD4 memory T cells were involved autoimmune processes, M1 macrophages functioned in proinflammatory, microbicidal, and tumor resistance processes, and resting dendritic cells induced tumor immune tolerance through the receipt of immunosuppression signals (*Adema, 2009*; *Cheng et al., 2019*; *Seledtsov & Seledtsova, 2019*). The distinct infiltration of immune cells implied that pancreatic cancers with

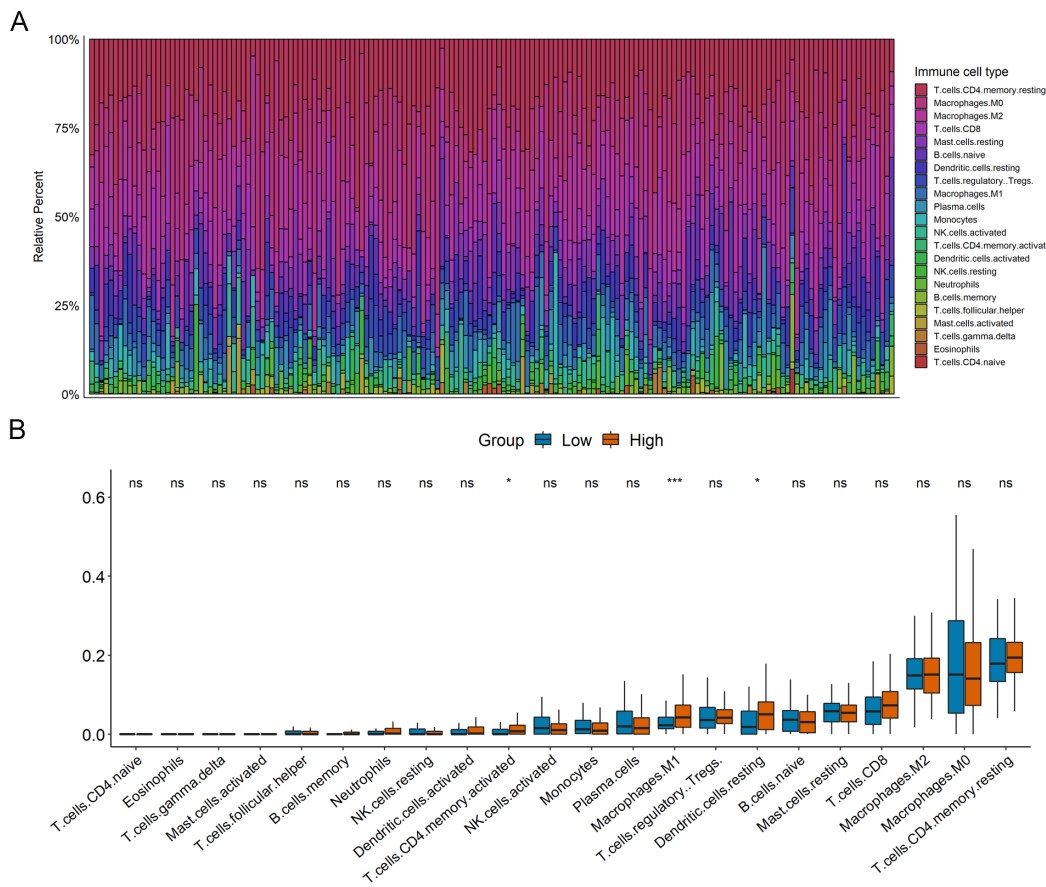

**Figure 5** **The relationship between GBP2 expression and immune infiltrating.** (A) The landscape of 22 subpopulations of immune cells in PAAD patients analyzed by CIBERSORT. (B) Comparison of immune cell components in high- and low-expression of GBP2 groups.

different GBP2 expressions may have different tumor immune microenvironments. We further compared the expression of common immune checkpoints between the high-GBP2 and low-GBP2 expression groups. In the overexpressed GBP2 group, immune checkpoint coding genes, including PDCD1, PDCD-L1, CTLA4, CD80, TIGIT, LAG3, IDO2 and VISTA, were significantly upregulated versus the low-GBP2 expression group. The IC molecules are important to the maintenance of immune homeostasis *in vivo*, and are involved in the tumor immune escape (*Karin, 2018*). The up-regulated expression of IC in the GBP2 high expression group may lead to the aggravation of immunosuppression, which may be a factor contributing to the poor prognosis in PAAD. Cancer immunotherapy has focused on blocking IC receptors and the expression level of PD-L1 has been showed to be an important predictor of immunotherapy (*Reck et al., 2016*). Therefore, the significant expression relationship between GBP2 and IC also hinted a question whether GBP2 has the value of prediction of immunotherapy response.

Previous studies have shown that GBP2 is induced by type I and type II IFNs, and have also reported that it is induced by IL-1β and TNF-α (*Klamp et al., 2003*; *Tripal et al.,*

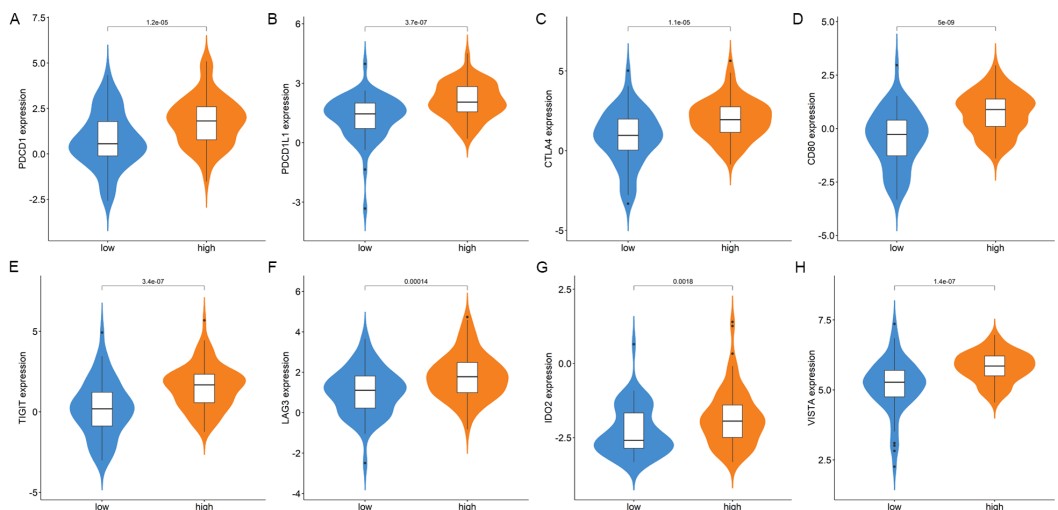

**Figure 6** **Comparison of the immune checkpoint expression in high- and low-expression of GBP2 groups.** (A) PDCD1. (B) PDCDL1. (C) CTLA4 (D) CD80. (E) TIGIT. (F) LAG3. (G) IDO2. (H) VISTA.

*2007*). GBP2 may logically be involved in host defense since a variety of viruses could lead to IFN production after invading the host. However, the occurrence of many malignant tumors is related to viral infection, including liver cancer and HBV, nasopharyngeal cancer and EBV, and cervical cancer and HPV. Whether the pathology of pancreatic cancer with high expression of GBP2 is also associated with viral infection is a question worthy of consideration and further exploration.

## CONCLUSIONS

In the present study, the expression of GBP2 was found to be elevated in PAAD tissues compared with the adjacent normal tissues. The high expression of GBP2 was closely related to a poor prognosis. The expression of GBP2 was an independent risk factor of prognosis and has a good predictive ability of for the 3-year survival in patients with PAAD. These results suggested that GBP2 may be a potential prognostic marker for PAAD. The significant correlation between GBP2 expression and immune infiltration, and ICs expression provide new insight for the future exploration of immunotherapy for PAAD.

## ACKNOWLEDGEMENTS

We thank our colleagues in the Department of Hepatobiliary Surgery for assisting us with our study.

### Funding

The project was funded by the Project of Chengdu Medical Research (2020118) and Scientific Research Program of Sichuan Provincial Health Commission (18PJ568). The

funders had no role in study design, data collection and analysis, decision to publish, or preparation of the manuscript.

## Grant Disclosures

The following grant information was disclosed by the authors:

Chengdu Medical Research: 2020118.

Scientific Research Program of Sichuan Provincial Health Commission: 18PJ568.

## Competing Interests

The authors declare there are no competing interests.

## Author Contributions

- Bo Liu performed the experiments, analyzed the data, prepared figures and/or tables, and approved the final draft.
- Rongfei Huang performed the experiments, authored or reviewed drafts of the paper, and approved the final draft.
- Tingting Fu analyzed the data, authored or reviewed drafts of the paper, and approved the final draft.
- Ping He analyzed the data, prepared figures and/or tables, and approved the final draft.
- Chengyou Du analyzed the data, prepared figures and/or tables, and approved the final draft.
- Wei Zhou analyzed the data, prepared figures and/or tables, and approved the final draft.
- Ke Xu conceived and designed the experiments, analyzed the data, authored or reviewed drafts of the paper, and approved the final draft.
- Tao Ren conceived and designed the experiments, authored or reviewed drafts of the paper, and approved the final draft.

## Human Ethics

The following information was supplied relating to ethical approvals (i.e., approving body and any reference numbers):

The Ethics Committee of The First Affiliated Hospital of Chengdu Medical College approved this research (2020CYFYIRB-BA-1200).

## DNA Deposition

The following information was supplied regarding the deposition of mRNA sequences:

The datasets were all downloaded from public databases: TCGA and GEO: GSE15471; GSE60979; GSE62165; GSE91035.

TCGA-PAAD: https://portal.gdc.cancer.gov/projects/TCGA-PAAD.

## Data Availability

The clinical information of the 42 pancreatic adenocarcinoma patients included in our cohort are available in a Supplemental File.

## Supplemental Information

Supplemental information for this article can be found online at http://dx.doi.org/10.7717/peerj.11423#supplemental-information.

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
