# Peer review of "GBP2 as a potential prognostic biomarker in pancreatic adenocarcinoma"

_PeerJ, doi:10.7717/peerj.11423_

## Round 0.1 · original submission · Major Revisions

Please provide a comprehensively revised version addressing the editorial comments and a detailed rebuttal letter. Also, check the comments in the annotated PDF file attached.

Reviewer 1 ·

Basic reporting

no comment

Experimental design

no comment

Validity of the findings

no comment

Additional comments

In this interesting work, authors showed a prognostic role of protein GBP2 in PAAD via assessment of the public data and the samples from their center. However, few issues need to be considered.
1, In the Abstract part, the Conclusion of this study is missing, and after the Discussion part, authors described the conclusion only use one sentence. This is against common practice.
2, Since GBPs play an important role in host defense against virus infection and the co-expressed genes of GBP2 were mainly associated with immune response, is there any possibility that the development of pancreatic cancer is related to virus infection?
3, In the Figure 2D, high expression of GBP2 is correlated to poor prognosis of PAAD. The representative diagrams of IHC corresponding to high and low expression of GBP2 should be showed.

Reviewer 2 ·

Basic reporting

The English language should be improved to ensure that an international audience can clearly understand your text. Some examples where the language could be improved include lines 49, 170-174, 222, 252-254, the current phrasing makes comprehension difficult.

Experimental design

No comment

Validity of the findings

1. I congratulate the authors for their study design, analytical integration favors a comprehensive analysis of the research topic. If there is a weakness, it is in the conclusion (concerning the inconsistency that I have noted in point 2) that it should be
reviewed before Acceptance.

2. Your analysis of microarray and TCGA-PAAD datasets has shown consistent results (Fig. 1B & 1C). However, these results are not concordant with your experimental results (Fig. 1E). Please explain this inconsistency.

3. Your discussion needs more detail. I suggest that you increase explanation about the implication of the relationship between GBP-2 and the immune checkpoints to improve your discussion.

Annotated reviews are not available for download in order to protect the identity of reviewers who chose to remain anonymous.

---

## Round 0.2 · accepted · Accept

Thanks for addressing all the revisions and corrections requested. Now your manuscript is accepted in PeerJ.

Reviewer 2 ·

Basic reporting

No comment

Experimental design

No comment

Validity of the findings

No comment